# Transcriptomic Complexity of Culm Growth and Development in Different Types of Moso Bamboo

**DOI:** 10.3390/ijms24087425

**Published:** 2023-04-18

**Authors:** Long Li, Binao Zhou, Dong Liu, Hongyu Wu, Qianqian Shi, Shuyan Lin, Wenjing Yao

**Affiliations:** 1Co-Innovation Center for Sustainable Forestry in Southern China, Nanjing Forestry University, Nanjing 210037, China; 2Bamboo Research Institute, Nanjing Forestry University, Nanjing 210037, China; 3College of Landscape Architecture and Art, Northwest A&F University, Xianyang 712100, China

**Keywords:** moso bamboo, growing culms, alternative splicing, lncRNA, isoform sequencing

## Abstract

Moso bamboo is capable of both sexual and asexual reproduction during natural growth, resulting in four distinct types of culms: the bamboo shoot-culm, the seedling stem, the leptomorph rhizome, and a long-ignored culm—the outward-rhizome. Sometimes, when the outward rhizomes break through the soil, they continue to grow longitudinally and develop into a new individual. However, the roles of alternative transcription start sites (aTSS) or termination sites (aTTS) as well as alternative splicing (AS) have not been comprehensively studied for their development. To re-annotate the moso bamboo genome and identify genome-wide aTSS, aTTS, and AS in growing culms, we utilized single-molecule long-read sequencing technology. In total, 169,433 non-redundant isoforms and 14,840 new gene loci were identified. Among 1311 lncRNAs, most of which showed a positive correlation with their target mRNAs, one-third of these IncRNAs were preferentially expressed in winter bamboo shoots. In addition, the predominant AS type observed in moso bamboo was intron retention, while aTSS and aTTS events occurred more frequently than AS. Notably, most genes with AS events were also accompanied by aTSS and aTTS events. Outward rhizome growth in moso bamboo was associated with a significant increase in intron retention, possibly due to changes in the growth environment. As different types of moso bamboo culms grow and develop, a significant number of isoforms undergo changes in their conserved domains due to the regulation of aTSS, aTTS, and AS. As a result, these isoforms may play different roles than their original functions. These isoforms then performed different functions from their original roles, contributing to the transcriptomic complexity of moso bamboo. Overall, this study provided a comprehensive overview of the transcriptomic changes underlying different types of moso bamboo culm growth and development.

## 1. Introduction

Moso bamboo (*Phyllostachys edulis*) is a versatile plant that plays a significant role in the food, papermaking, and construction industries, making it one of the most valuable non-timber forestry resources in East Asia. With one of the fastest growth rates in the plant kingdom, Moso bamboo exhibits four distinct culm growth patterns under natural settings: bamboo shoot, outward-rhizome, seedling stem, and leptomorph rhizome.

Moso bamboo propagates through both sexual and asexual reproduction. It can develop new individuals from a seed, although the first culm will grow to about 15 cm high and the growth rate of these seedlings is very slow. Moreover, the flowering and fruiting of moso bamboo are rare. After flowering, bamboo typically enters a period of decline in growth and may eventually die [1]. In many cases, bamboo reproduces asexually by developing new individuals from lateral buds located on the basal part of the rhizome internode, which ultimately develop into new shoots. Based on their growth characteristics, the growth process of bamboo shoots can be divided into the winter, early, and late periods [2]. In winter, the bamboo shoot undergoes primary thickening growth for about six months, starting in September and ending in March [3]. In spring, the intercalary meristem of bamboo shoots shows vigorous growth in the early period, followed by a decline [4]. In a mature bamboo forest, a shoot can grow up to 1 m per day in suitable spring conditions and may reach a final height of 15–20 m in just 1.5 to 2 months [1]. The vigorously growing shoots of bamboo absorb nutrients and energy from the robust rhizome-root system, which can extend horizontally and connect the young shoots (culms) with other mature bamboos [5]. The outward rhizomes are induced by many environmental factors. When stones or other rhizomes block rhizome extension, the tips of the rhizome pierce through the ground and develop into a small bamboo plant [6].

Alternative pre-mRNA splicing (AS) is a ubiquitous post-transcriptional mechanism for controlling isoform expression in higher eukaryotes, which enhances the diversity of both the transcriptome and proteome [7,8]. There are five main types of AS events, including alternative 5′ splice sites (3′SS), alternative 5′ splice sites (5′SS), intron retention (IR), exon skipping (ES), and mutually exclusive exons (MXE). Nearly 95% genes undergo AS in human genome, and exon skipping is predominant type. By contrast, intron retention is the dominant AS type in plants. The alternative transcription start or termination site (aTSS or aTTS) could assist in post-transcriptional regulation. It affects protein translation efficiency and mRNA stability [9], which are significantly different from the regulatory role of AS. It is evident that both pre-transcriptional regulation and post-transcriptional modifications have been involved in numerous physiological and adaptive regulations. Moreover, AS is closely associated with plant growth and development processes [10], as well as environmental adaptability [11].

Many studies have reported the general growth mode, underground growth stage [12], growth and development dynamics [13], pith cavity formation [14], transverse development [15], and transcriptomic changes [2,16] in bamboo shoots. Despite reports of identification of the AS events in the moso bamboo genome [17,18], it remains unexplored how the combined action of AS, aTSS, and aTTS has contributed to transcriptome complexity in different types of growing culms. How the diverse transcripts respond to developmental cues and what relevance these outcomes have to the acquisition or loss of conserved domains, as well as their impact on gene function diversity, remain unclear.

Long non-coding RNAs (lncRNAs) are defined as transcripts longer than 200 bp that cannot encode a full-length protein, and the main difference between lncRNA and mRNA is the lack of coding protein sequence potential in lncRNA [19,20,21]. Recently, more and more evidence has suggested that lncRNAs have regulatory roles in various biological processes, such as development, vernalization, and environmental stress adaptation. For example, long noncoding RNAs and microRNAs have provided valuable insight into gene expression regulation in the bread wheat response to drought stress [22]. Furthermore, many wheat LncRNAs carry conserved microRNA precursors, and LncRNAs collaborate with microRNA and form complex regulatory networks in insect responses [23].

Compared to second-generation sequencing technology (SGS), single-molecule long-read sequencing (SLS) technology significantly improves read length [24] and avoids transcriptome assembly. SLS technology has facilitated the discovery of novel gene loci, transcripts, and AS events in various plants species, including *Sorghum bicolor*, *Aethionema arabicum*, *Gossypium hirsutum*, and *Populus* [17,25,26,27]. However, studies on the moso bamboo rhizome-root system have gone further, revealing previously unidentified gene models, lncRNA, and AS events [28]. Neverthless, further investigation is required to identify isoforms, AS, aTSS, and aTTS in other parts of the bamboo, such as the seedling stem, outward rhizome, and bamboo shoot.

This study identified genome-wide isoforms across four distinct culm types using the PacBio Isoform-Sequencing technology. Subsequently, the analysis focused on AS, aTSS, and aTTS to investigate their potential roles in gene regulation and uncover transcriptomic complexity within the different culms. In addition, we elucidated the cause of the outward rhizome formation from the perspective of AS, aTSS, and aTTS. This study will uncover novel avenues for investigating the regulatory roles of AS, aTTS, and aTSS in modulating conserved domains that are critical for protein function in different types of growing culms.

## 2. Results

### 2.1. Isoform Detection and Characterization in Moso Bamboo Growing Culms

To improve the current annotations and identify the AS events, aTSS, and aTTS in different types of culms growth, we sequenced the transcriptome of different types of growing culms using the Pacific Biosciences Isoform Sequencing platform, which generates long reads that can span entire transcripts. Two isoform sequencing libraries were constructed using RNA mixes from underground tissues, including winter bamboo shoots (B1), rhizomes (R), and lateral buds (L), and from aboveground growing tissues, including spring bamboo shoots (B2), outward rhizomes (OR), and seedling stems at heights of 1.5 cm (S1) and 4.5 cm (S2), respectively (Appendix A). A total of 1023,811 FLNC reads were obtained. After SGS data correction (Appendix A), the resulting set of high-quality consensus transcript sequences and error-corrected FLNC reads were mapped to the Moso bamboo genome, yielding 169,433 non-redundant isoforms. Of these, 50,936 isoforms were known and mapped to the moso bamboo genome annotation database, while 97,289 were new isoforms derived from annotated gene sets. Additionally, we identified 21,208 novel isoforms from 14,840 previously unannotated gene loci (Figure 1A, Appendix A). Among the 65,776 moso bamboo genes, 6949 were mono-exon genes, 13,672 contained two exons, and the number of genes gradually decreased with an increase in the number of exons per gene.

Of all the genes detected by Iso-Seq, 27,244 contained two or more isoforms, indicating that these genes underwent AS, aTSS, or aTTS events. Among these 27,244 genes, 9092 had two isoforms, while only 2113 had more than 10 isoforms. The number of both newly identified genes and annotated genes shows a decreasing trend as the number of isoforms per gene increases. Furthermore, the annotated genes in the moso bamboo genome database had more isoforms than the newly identified genes, and more than half of the novelly identified genes contained only one isoform (Figure 1C). When examining expressed isoforms in different growing culm tissues, only 55,111 isoforms were found to be expressed in all seven samples (Figure 1D), while 8429 isoforms were specifically expressed in just one tissue. Notably, 1934 isoforms were specifically expressed in B1, which was much higher than in any other samples. To sum up, a large number of isoforms are specifically expressed in one or more tissues, and these isoforms that are specifically expressed in tissues may be regulated by AS, aTSS, or aTTS events, which requires further study. 

In order to explore relationships among different growth models of Moso bamboo culms, we performed principal component analysis (PCA) on the isoform expression dataset (Figure 1E). This allowed us to visually display transcriptional signatures and developmental similarities. The first component separated the outward rhizome from the other six bamboo samples, while the second component separated the two seedling stem samples from other samples obtained from bamboo forests. In addition, within group III, the three underground samples (B1, R, and L) showed a much closer relationship to each other than to B2.

In our full-length transcripts, we found substantial differences between the previous annotation project [29] and our full-length transcripts, which identified 50,396 annotated genes in the moso bamboo genome database (AGs) and 14,840 novel unannotated genes (UAGs). First, we found that the AGs had significantly higher GC content than the UAGs based on the Wilcoxon rank sum test (Figure 2A), indicating that more genes with lower GC content were missed in the moso bamboo genome annotation project. Second, a comparison of gene numbers of mono- and multi-exon genes among AGs and UAGs indicated a higher percentage of mono-exon genes being missed in the previous annotation (Figure 2B). Third, the median values of mis-annotated genes were much higher compared to UAGs, making it more difficult to reconstruct using transcriptome assembly by SGS technologies (Figure 2C).

### 2.2. Profiling of Global Poly(A) Sites and lncRNAs

Of the 65,776 genes detected by Iso-Seq, 18,348 genes had at least one poly(A) site, and 3103 genes had only one poly(A) site (Figure 3A). The number of genes gradually decreased as the number of poly(A) sites per gene increased, and only 2593 genes had more than six poly(A) sites. We performed a MEME-ChIP analysis for motifs enriched upstream of the cleavage site. Three significantly over-represented polyadenylation signals that harbored at 50 nucleotides upstream from the predominant poly(A) site of all expressed transcripts were identified, including TCTGT, CTTGTG, and CGGGGG (Figure 3B). Additionally, the nucleotide composition around the poly(A) cleavage sites for nucleotide bias was analyzed. There was a clear preference for T across the entire region, except for the −25 to −10 and 0 to 10 regions, which were predominant for A, suggesting a nucleotide bias upstream and downstream of the cleavage site in 3′ UTRs (Figure 3C). The nucleotide bias in these sites was similar to that previously reported [20], which supports the authenticity of the identified poly(A) sites.

LncRNAs are involved in various biological processes in both plants and animals, but the investigation of the whole-genome lncRNAs involved in different types of moso bamboo culm growth is lacking. Four computational methods were used to identify lncRNAs from the 169,433 isoforms obtained by Iso-Seq (Figure 4A). The length of most lncRNAs was 500–4000 bp (Figure 4B). Among these, 292 lncRNAs with a length between 2500 and 3000 bp, which occupied the largest portion, 261 lncRNAs with a length between 1500 and 2000 bp. The average length of all identified 1ncRNAs is 2259.64 bp, much longer than the known length of lncRNAs (1284 bp). The lncRNAs contained an average of 1.83 exons, while the mRNAs contained an average of 7.45 exons.

In order to explore the potential roles of lncRNA involved in different types of moso bamboo culm growth and development, we analyzed their expression pattern. 1311 expressed lncRNAs could gather into five clusters (Figure 4C). Cluster 1, accounting for one-fourth of all lncRNAs, was highly expressed in S1 and S2. The lncRNAs in clusters 2 and 3, which accounted for one third of the total lncRNAs, presented expression preferences in B2, OR, or both. Most lncRNAs in cluster 4 showed high expression levels in B1 and R. The lncRNAs in cluster 5 displayed high accumulation levels in R, L, or both. Furthermore, 363 lncRNAs showed the highest accumulation level in B1. GO enrichment analysis demonstrated that genes involved in several biological networks, including anatomical structure development (GO:0048856), multicellular organism development (GO:0007275), reproduction (GO:0000003), cell cycle (GO:0007049), post-embryonic development (GO:0009791), et al., were significantly affected by these lncRNAs (Figure 4C). We further assessed the expression correlation between lncRNAs and their adjacent (for lincRNAs) or anti-strand (for lncNATs). The result suggested that 4931 co-expression pairs of lncRNAs and mRNAs showed a significant positive correlation (Pearson correlation coefficient > 0.6, PCCs), while 2650 showed a significant negative correlation (PCCs < −0.6) (Appendix A). Furthermore, nearly one-third of the lncRNAs showed the highest accumulation level in B1, indicating the importance of lncRNAs involved in moso bamboo shoot growth by affecting biological processes such as anatomical structure development, multicellular organism development, reproduction, cell cycle, et al.

### 2.3. AS, aTTS, and aTSS Increased Transcriptome Complexity in Moso Bamboo Growing Culms

A total of 102,315 AS events were identified in the moso bamboo genome, which impacted 14,923 moso bamboo genes (Appendix A). Among various types of moso bamboo growing culms, the most prevalent AS event was intron retention, which accounted for 27% of all alternatively spliced events (Figure 5A). Contrarily, exon skipping occurred in only 5% of cases. Moreover, the use of alternative 3′ splice sites (8%) was more frequent than the use of alternative 5′ splice sites (5%). 

To identify the factors influencing alternative splicing, we analyzed the characteristics of alternative 5′ and 3′ splice sites and computed correlations between the frequency of alternative splicing and factors such as exon length, exon number, mRNA length, and GC content. The occurrence of alternative 5′ and 3′ splice sites was mainly located within 4 nucleotides upstream and downstream of the predominant splice sites (Figure 5B and Figure 5C, respectively). Notably, these activated splicing sites were found to be still associated with GU and AG dinucleotides, thereby suggesting that this AS pattern was conserved in eukaryotes (Figure 5D,E). With the increase of exon numbers and mRNA length, AS enhanced transcript diversity and complexity in growing culms of moso bamboo (Figure 6A and Figure 6B, respectively). Conversely, exon length and GC content showed a negative correlation with AS frequency(Figure 6C and Figure 6D, respectively). These results indicated a strong dependence of AS occurrence on genic or genomic features.

To validate AS events identified by SLS, we performed SqRT-PCR on three genes that exhibited a total of four AS events (Figure 5F). The SqRT-PCR results were consistent with our expectations and validated the reliability of AS events.

We compared the isoform number between paralogous gene pairs located in collinear regions and finally identified a total of 21,496 paralogous gene pairs (Figure 7A). Among these gene pairs, 9814 contained more than two isoforms in both two paralogous genes, while 4452 contained only one isoform. These results suggest that there is no obvious correlation between the number of isoforms and the presence of multiple isoforms in its paralogous genes located in collinear regions.

The diversity of isoforms is collectively contributed by alternative promoter selection, alternative termination sites, and selective splicing of introns, which also play significant roles in regulating gene expression. In order to explore potential correlations among these three types of events, we performed a Venn analysis. Statistical analysis revealed that 26,844 genes were affected by both aTSS and aTTS processes, while few genes were affected only by aTSS or aTTS processes (Figure 7B). The vast majority of genes coupled with the AS event were also accompanied by aTSS and aTTS. We extended the differentially expressed genes analysis to include all pairwise comparisons of the 7 samples and identified 34,319 DEGs based on the defined criteria. For all genes where isoform-generating events occurred, 27,173 genes were differentially expressed, accounting for 79.5% of all DEGs. Among these, 12,377 differentially expressed genes were accompanied by AS, and 14,722 differentially expressed genes were accompanied by both aTSS and aTTS. These results suggested that most AS-associated genes were also accompanied by aTSS and aTTS.

### 2.4. AS, ATSS, and ATTS Are Indispensable Regulation Approaches Regulating Different Types of Moso Bamboo Culm Growth

As a special growth mode of bamboo culms, the growth pattern of outward rhizomes was different from that of the bamboo shoot and rhizome. When underground rhizome growth is obstructed by obstacles such as stones and other rhizomes, they continue to grow longitudinally, and new individuals form above ground. Thus, we investigated whether AS affected the outward rhizome growth and development by comparing the AS events between the outward rhizome and the winter bamboo shoot, spring bamboo shoot, rhizome, and lateral bud. The results showed that a large number of genes were differentially alternatively spliced during outward rhizome growth and development when compared with the other four samples. In all four pairwise comparison groups, most differentially AS events were related to intron retention, and most of these are enhanced in the outward rhizome (Figure 8A). Next, we categorized the GO function of these DASs, and the GO terms, including cell cycle, DNA metabolic process, response to stress, metabolic process, et al., were most significantly enriched in all four comparison groups (Figure 8B). These results indicated that AS events, especially intron retention, significantly affected the cell cycle, DNA metabolic process, and response to stress-associated isoform expression, eventually exerting a positive or negative influence on outward-rhizome growth and development.

Although the investigation of genome-wide splice junction selection in moso bamboo has recently been conducted using Illumina or PacBio reads [23], the analysis of the relative contribution and functional effect of biological processes has been limited. To assess the influence that AS, ATSS, and ATTS exert on domain gain and loss, which lead to functional changes in these important biological processes, we selected four plant growth and development-associated families, including cyclin A, NAC, E2F, and GRF (growth-regulating factor), for further conserved domain and expression pattern analysis at the isoform level. Transcript isoforms were translated based on the longest ORF by CPC2. Generally speaking, most isoforms that suffered domain loss showed a much lower expression level than isoforms that contained all integrated domains (Figure 9). However, there were some differences in tissue expression specificity among these isoforms. In the Cyclin A family, most isoforms, including six incomplete isoforms that lost the cyclin domain, showed the highest expression level in the winter bamboo shoot. One isoform with domain loss was preferably expressed in the rhizome. The other five isoforms that lost one or both cyclin domains showed low expression abundance in all seven tissues. In the NAC family, 11 isoforms lack the UBA/TS domain at their C-terminal. Seven of these isoforms were predominantly expressed in underground tissues (B1, R and L), while the other four isoforms, such as PB.5756.1, PB.21657.2, PB.6925.4 and PH02Gene0907.t1, exhibited high accumulation levels in spring bamboo shoots. In the E2F family, six isoforms underwent domain loss, acquisition, or both. Of these, two obtained novel functional domains—peroxidase, while another two switched their original domain for a new functional one. These four isoforms showed a high expression level in the outward rhizome. In the GRF family, nearly half of the isoforms performed at their highest expression levels in the winter bamboo shoot, of which only three isoforms suffered domain loss or domain acquisition events. Similarly, only six isoforms showed the highest expression levels in spring bamboo shoots, but four of these isoforms underwent domain loss or acquisition events.

## 3. Discussion

Usually, AS events are commonly classified into intron retention, exon skipping, the alternative 5′ splice site, the alternative 3′ splice site, and mutually exclusive exons, and the occurrence of these changes could impact the fate of gene products. Alternative transcription initiation and alternative transcription termination are also important transcriptional regulation processes that contribute to transcript complexity in plants [10]. Based on single molecular long-read sequencing data, we identified that 27,244 genes produced at least two isoforms in growing culms, suggesting the vital function of AS, aTSS, and aTTS in moso bamboo culm growth. We compared the isoform number between paralogous gene pairs locating in the collinear region and found that a gene raised from AS, ATS, or ATT did not mean that its paralogous genes were also raised from these events. Although the large-scale genome duplications events caused a large number of paralogous gene pairs to harbor in the collinear region [1], the pre-transcriptional and transcriptional regulation roles of these paralogous pairs seemed to alter during evolution. Intriguingly, most AS-occurring genes were also accompanied by aTSS and aTTS. Furthermore, the frequencies of aTSS and aTTS were more common than that of AS, which was consistent with the observations in human genes [30] and also similar to the response to fruit development and maturation of strawberry genes [10].

In pairwise comparison, the most significantly abundant AS events in the outward rhizome were associated with intron retention. Previous studies suggested that stress-induced AS events were characterized by increased intron retention in plants [31]. As rhizomes grow underground, they may form new individuals (outward rhizomes) by breaking through the soil when their extension is blocked by obstacles [6], suggesting that the change in growth pattern forced them to adapt to a new environment and that it may also cause stress for rhizomes growing to some extent. This could partially explain the significant increase in intron retention events observed in outward rhizomes.

AS, aTSS, and aTTS affects protein functions in various ways, such as altering amino acid composition, secondary structure stability, conserved domain modification, and combining ability. Domains can be regarded as distinct functional and structural units of a protein [32], and the domain loss and acquisition also associated with protein function change. In this study, a large amount of plant growth and development associated genes lost their functional domains, which were caused by these three types of events regulation. Cyclin domains were considered as core elements of cyclin families and played crucial roles in controlling the cell progression via activating cyclin-dependent kinase enzymes [33]. Five out of six non-domain isoforms showed much higher concentrations in rhizome than others, indicating that AS event might exert a certain extent of negative influence on rhizome growth. Unlike domain loss occupying the dominant position in Cyclin A and NAC families, the domain acquisition occupied much larger proportion in E2F and GRF gene families than domain loss. Four E2F isoforms were inserted novel functional domain, and all of them preferably expressed in outward rhizome. It was reported that the protein with peroxidases domain were found in the extracellular space or in the vacuole, where they have been implicated in hydrogen peroxide detoxification, lignin biosynthesis, and stress response [34]. Thus, two E2F isoforms obtained peroxidases domain might be caused by environmental change cue that associated with stress response, and partially contributed to lignin biosynthesis. In GRF families, five isoforms obtained F-box-like domain. The F-box domain was commonly found at the N-terminus of various proteins and played a critical role in plant response to various biotic and abiotic stresses via mediating protein–protein interactions in a variety of contexts, such as polyubiquitination, transcription elongation, centromere binding and translational repression [35]. Thus, the newly obtained F-box-like domain helped these isoforms improve new functions including plant stress tolerance. Furthermore, the GRFs regulated plant growth such as leaf expansion [36], root development [37], stem elongation [36], and tillering [38]. Thus, a large amount of GRFs harboring both complete WRC and QLQ domain showed high accumulation level in winter bamboo shoot, and were essential for winter bamboo shoot growth.

LncRNAs exhibit diverse features and regulatory mechanisms in moso bamboo growing culms. Consistent with previous work in other plants, we also observed a high degree of tissue specificity among lncRNAs [21,39]. Of the 1311 lncRNAs we identified, most of the co-expression pairs with mRNAs were positively correlated, and one-third of them were preferentially expressed in winter bamboo shoot [40]. Moreover, it has been documented that lncRNAs promoted plant growth, regulate plant life activities, and response to various biotic or abotic stress, such as seed germination [41], lateral root formation [42], fiber accumulation [43], lipid metabolism [44], leaf development [45], drought response [22], and insect response [23]. Based on lncRNA expression and GO annotation of their target genes, we deduced that their high accumulation level in winter bamboo shoot promoted anatomical structure development, multicellular organism development, reproduction, cell cycle, and other plant growth and development associated gene transcription, and exerted a positive effect on bamboo shoot growth.

Unlike previous reports on animals and other model plants [46,47], lncRNAs identified in the current Iso-Seq data of moso bamboo growing culms were notably longer. The median length of these lncRNAs was more than twice that of previously described lncRNAs, which may be attributed to their low expression, thus making it difficult for short reads assembly by second generation sequencing.

## 4. Materials and Methods

### 4.1. Sample Collection and RNA Extraction

Except seedling stems, the tissues of different types of moso bamboo growing culms were collected from Xuancheng (E119°41′; N30°89′), Anhui Province. Bamboo seeds were germinated and grown in phytotron. Seven represented tissues including winter bamboo shoot (B1), 1.5 m height spring bamboo shoot (B2), lateral buds (L), 0.5 m long rhizomes (R), 0.3 m long outward rhizomes (OR), 1.5 cm height seedling stem (S1), and 4.5 cm height seedling stem (S2) were selected. B2, OR, S1, and R were sampled when the culms reached about one tenth of the final length. For each culm sample, the top fifteen percent length of the culms were defined as the culm tips and used for further transcriptome sequencing (Figure 1).

Total RNA from each sample was extracted from each culm samples using TRIzol reagent (Invitrogen, Carlsbad, CA, USA). RNA with high purity and integrity which examined by Agilent 2100 TapeStation system (Agilent Technologies, Santa Clara, CA, USA) and Nano Photometer spectrophotometer (IMPLEN, Munich, Germany) was further used for the library construction.

### 4.2. PacBio Single-Molecule Long-Read Sequencing Sequencing and Analysis

Two Pacific Biosciences single-molecule long-read sequencing libraries were constructed based on equally combined total RNA from underground samples (B1, R, and L) and aboveground tissues (B2, OR, S1, and S2), respectively. After PCR amplification, the cDNA products were applied to create the SMRT bell Template library according to Pacific Biosciences Iso-Seq instructions (PacBio, Menlo Park, CA, USA). The iso-Seq libraries were sequenced by a PacBio RS II instrument (PacBio, CA, USA) with three SMRT cells.

The Iso-Seq data were further analysed by SMRTlink4.0 through a standard Iso-Seq protocol. Firstly, reads of inserts (ROIs) were generated from subreads removing adapters and artefacts, and then reads were classified into full-length and non-full-length reads using ‘pbclassify.py’ with default settings [48]. After the isoform level cluster, high-quality (post-correction accuracy above 99% quality value ≥ 30) consensus reads were obtained. Finally, high quality isoforms were aligned to the reference genome using GMAP2 [48]. The transcript structures were generated using TAPIS pipeline (version 1.0).

### 4.3. RNA-Sequencing Library Preparation, Sequencing and Analysis

Strand-specific RNA-Seq library was construct using the Next Ultra II RNA Library Preparation Kit (NEB, Moline, MS, USA), and sequenced using an Illumina HiSeq X Ten platform (San Diego, CA, USA), subsequently. The PacBio Iso-Seq and RNA-Seq datasets were deposited at the NCBI under accession numbers PRJNA706151 and PRJNA604634. The clean reads were then mapped to the reference transcript models built by Iso-Seq mentioned above [29]. The FPKM value (fragments per kilobase of transcript per million mapped reads) was used for the normalization of isoform expression levels.

### 4.4. Functional Annotation of Genes and Isoforms

Annotations of all genes and isoforms were based on BLASTX hits in four public databases including Gene Ontology (GO)(version version 2.2), KEGG (Kyoto Encyclopedia of Genes and Genomes), Kyoto Encyclopedia of Genes and Genomes (KEGG)( version version 3.10), Swiss-Prot (version 35), and Pfam protein domain (version 31.0) (E-value ≤ 10^−5^). GO enrichment was tested using topGO software (version 2.36.0).

### 4.5. Alternative Splicing Events

Five major AS event types, including intron retention, exon skipping, alternative 5′ splice site, alternative 3′ splice site, and mutually exclusive exons, were identified by AStalavista v4.0.1 according to the output files [49]. Differentially alternative splicing events (DAS) between different samples were respectively quantified using rMATS (version 4.1.2) [50]. Transcript isoforms were translated based on the longest ORF by TBtools v1.108. The conserved domain of each isoform was identified by Pfam protein domain (version 31.0).

### 4.6. Verification of AS Events

Six AS events were validated by RT-PCR using a set of primers that were designed based on each AS event. First-strand cDNA synthesis was conducted with approximately 1 μg of RNA using the reverse transcriptase AMV (Promega, Madison, WI, USA). The primers were designed by Oligo v7.56 software. After PCR amplification, The PCR products were isolated and cloned onto pMD18-T vector (Promega, Madison, Wisconsin, USA) for sequencing via ABI 3100 automated sequencer (Invitrogen, Carlsbad, CA, USA).

### 4.7. Poly(A) Analysis

To identify poly(A) tails, pacific Biosciences’ SMRT v1.2 analysis pipeline was employed using the full-length reads. The poly(A) sites were determined when there were at least eight adenine (A) bases and less than two non-A bases in 30 bases for all full-length reads. The motif peaks on the sequence of 50 nucleotides upstream of the poly(A) sites were visulized using MEME-ChIP tool (version 5.5.2) which run locally on a Linux system.

### 4.8. Identification of Long Intergenic Noncoding RNAs

The full-length reads which were not aligned to the known gene models obtained from moso bamboo genome annotation database, were identified as novel transcripts. Transcripts encoding ORFs longer than 100 amino acids yet has a nucleotide sequence less than 350 were filtered, and the remaining transcripts were further screened by The CPAT v1.2.4 and PLEK v1.2.

## 5. Conclusions

In this study, we have systematically investigated the transcriptome complexity using single molecular long-read sequencing techniques along the four different types of growing bamboo culms. The isoform sequencing results unraveled thousands of previously unexplored transcript isoforms. The lncRNAs played important roles in moso bamboo shoot growth by positively regulating target mRNAs expression. Additionally, most genes with AS were also accompanied by aTSS and aTTS. The predominant type of AS in moso bamboo was intron retention, which significantly increased in the outward rhizome due to the change in growth environment from underground to aboveground. Conserved domain analysis indicated that aTSS, aTTS, and AS significantly affected protein function by regulating the lost or acquisition of conserved domains. Overall, our results provide a new comprehensive overview of the dynamic transcriptomic landscape in different types of moso bamboo culm growth and development.

## Figures and Tables

**Figure 1 ijms-24-07425-f001:**
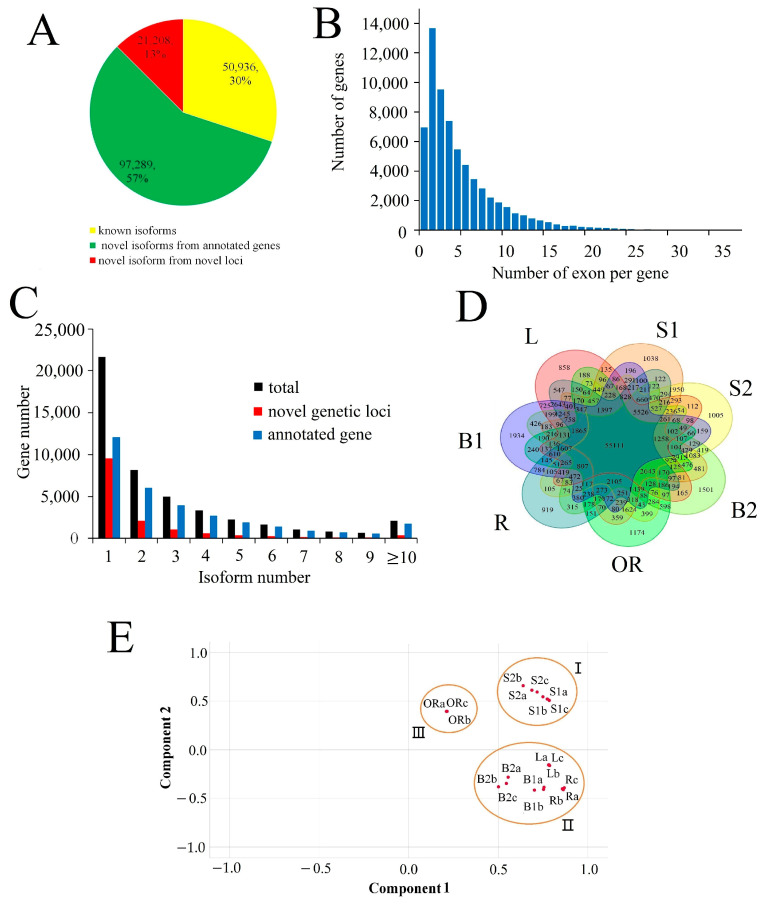
Summary of single molecular long-read sequencing data. (**A**) Categories of isoforms identified by single molecular long-read sequencing compared to the Moso bamboo genome annotation database. (**B**) Number of exons per gene for the isoform sequencing data. (**C**) Distribution of the splice isoforms per gene. The blue column, red column, and black column represented annotated genes, novelly identified genes, and all genes, respectively. (**D**) Venn diagram showing the overlapping sets of specifically expressed isoforms in seven different tissues. (**E**) Principal component analysis based on isoform expression level shows three distinct groups (marked as I, II, and III).

**Figure 2 ijms-24-07425-f002:**
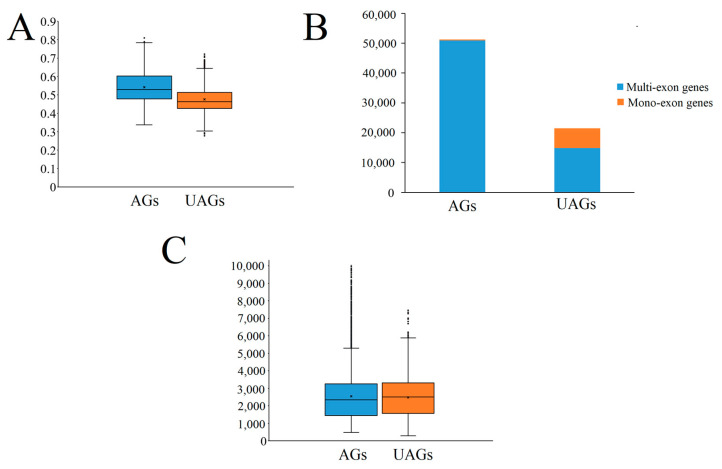
Comparison of the novel genes identified by isoform sequencing and previous annotated genes. (**A**) Distribution of GC content among the AGs and UAGs. (**B**) Comparison of gene numbers of mono-exon and multi-exon genes among AGs and UAGs. (**C**) Comparison of mRNA length between AGs and UAGs.

**Figure 3 ijms-24-07425-f003:**
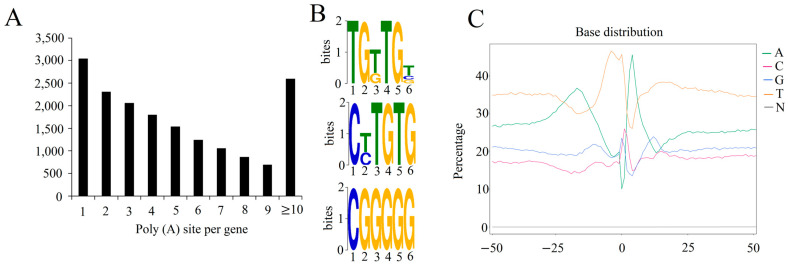
Features of alternative poly(A) sites in Moso bamboo growing culms. (**A**) Quantity of poly(A) sites per gene. (**B**) Three typical poly(A) signals identified by MEME in the moso bamboo genome. (**C**) The nucleotide composition profile around poly(A) cleavage sites.

**Figure 4 ijms-24-07425-f004:**
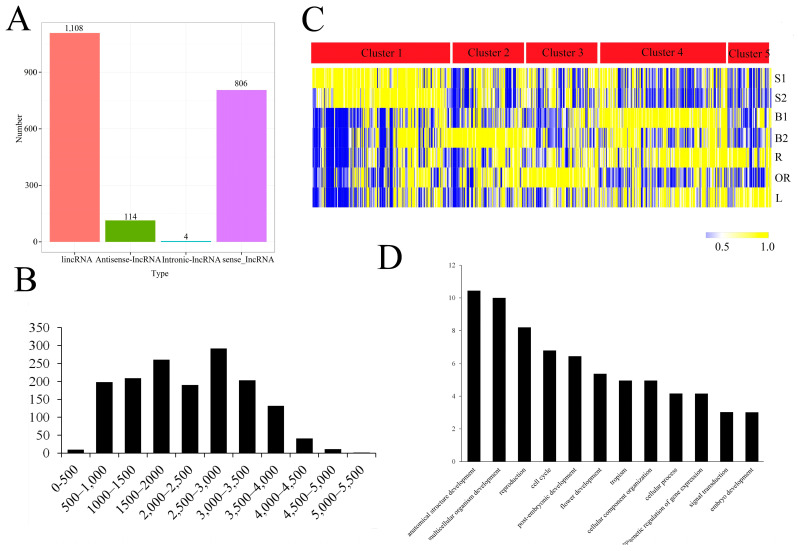
Features of long non-coding RNAs in the Moso bamboo growing culms. (**A**) The quantity of each type of lncRNA. (**B**) The lncRNA length distribution. (**C**) Expression patterns of all 1311 expressed lncRNAs. (**D**) GO annotation of all predicted target genes of lncRNA.

**Figure 5 ijms-24-07425-f005:**
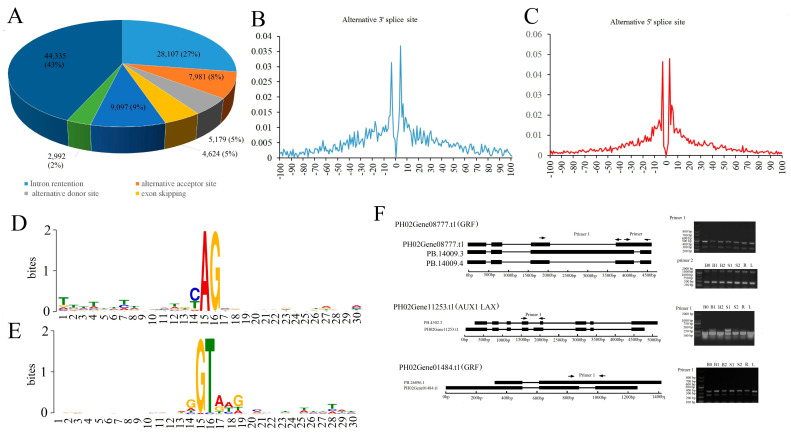
Characteristics of alternative splicing events in Moso bamboo growing culms. (**A**) Statistics of the different types of AS events in the Moso bamboo genome. (**B**,**C**) The distribution of the activated 5′ splicing sites (**B**) and the activated 3′ acceptor sites (**C**) around the dominant splice sites. (**D**,**E**) The nucleotide sequences around the alternative 5′ donor sites (**D**) and 3′ splicing sites (**E**). (**F**) PCR validation of six AS events identified by Iso-Seq.

**Figure 6 ijms-24-07425-f006:**
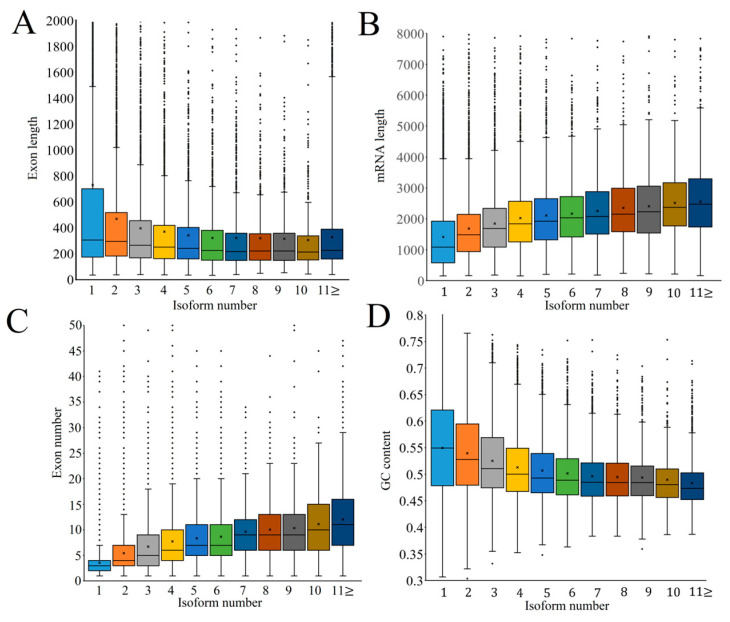
The relationship between gene structures and the number of isoforms per gene. (**A**) Exon length. (**B**) mRNA. (**C**) Exon number. (**D**) GC content.

**Figure 7 ijms-24-07425-f007:**
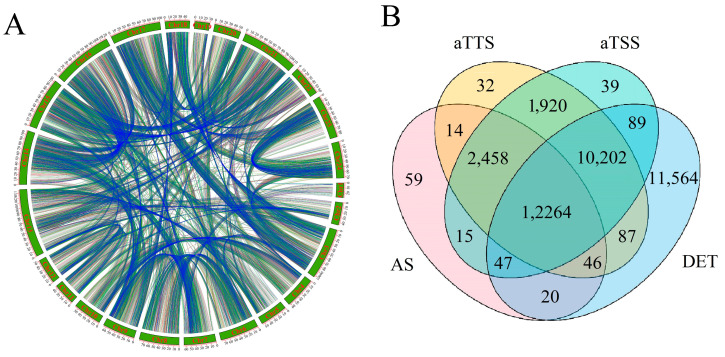
(**A**) Comparison of isoform numbers between paralogous gene pairs located in collinear regions. The two paralogous pairs connected by the blue line indicate that both genes contain more than two isoforms. The two paralogous pairs connected by the red line indicate that both genes contain only one isoform. The two paralogous pairs connected by a green line indicate that one gene contains only one isoform, while the other contains multiple isoforms. (**B**) The correlation of alternative splicing (AS), alternative transcript start sites (aTSS), and alternative transcript terminal sites (aTTS). The Venn diagram shows the overlapping genes subject to aTSS, aTTS, AS, or differential expression.

**Figure 8 ijms-24-07425-f008:**
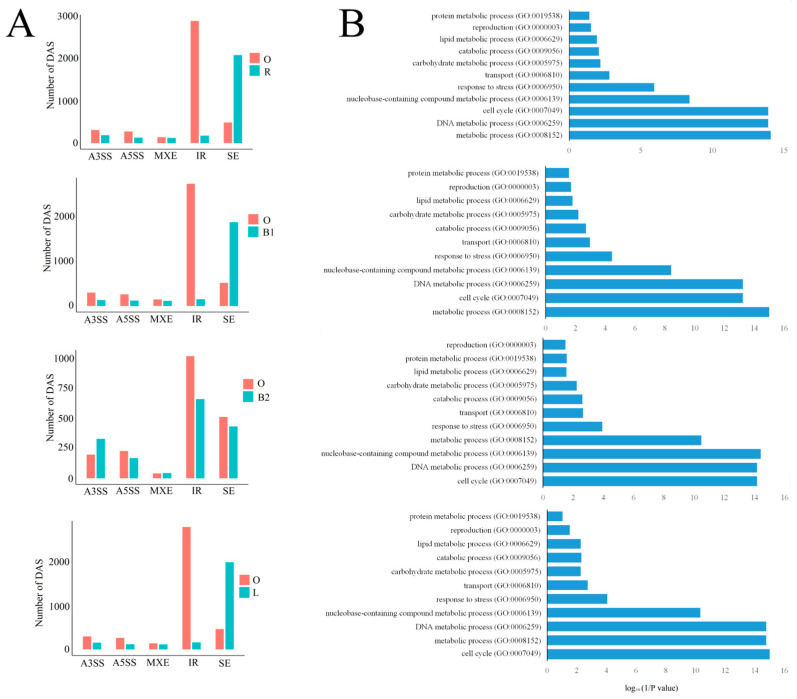
(**A**) The differentially alternative splicing events in the pairwise comparison. Genes derived from alternative 3′ acceptor site (A3SS), alternative 5′ donor site (A5SS), exon skipping (ES), intron retention (IR), and mutually exclusive exon (MXE) were summarized separately in the rMATs package. (**B**) Functional enrichment (biological process) of differential AS genes.

**Figure 9 ijms-24-07425-f009:**
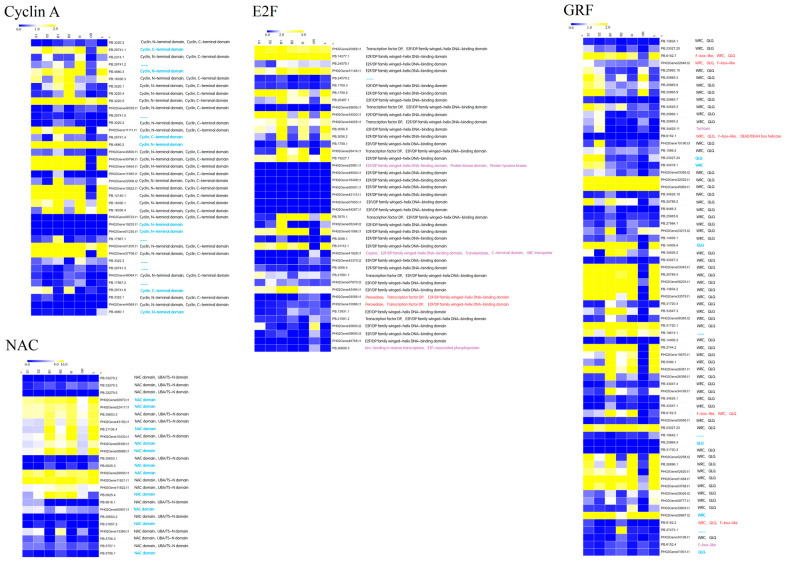
Expression and conserved domain analysis of isoforms from four important plant growth and development families. The color scale represented log2-transformed FPKM values. Yellow indicated high expression and blue indicated low expression. For each isoform, the identified conserved domains were shown following the isoform names. Blue or red represented the isoform that suffered from domain loss or domain acquisition, respectively. Purple represents the domain loss and domain acquisition that simultaneously occurred in that isoform.

## Data Availability

Raw sequencing data can be obtained from the NCBI under accession numbers PRJNA706151 and PRJNA604634. All the pertinent data are presented in the manuscript and associated Appendix A.

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
