# Peer review of "Transcriptomic Complexity of Culm Growth and Development in Different Types of Moso Bamboo"

_ijms, 2023, doi:10.3390/ijms24087425_

Round 1

Author Response

The manuscript submitted by Long et al. studied the transcriptome complexities on four different types of Moso bamboo culms during growth and development. The work was carried out well and the analysis were performed thoroughly, the results are interesting. The work represents a significant contribution to the plant community. My only concern in data analysis is the method used for non-coding RNA GO functional analysis (Figure 4d). As it is stated that GO analysis was based on BLASTX, non-coding RNAs do not code, thus, there should be no BLASTX hit expected for these sequences. If there is a hit, the translation product is not real. Thus, it is questionable for using this approach for functional analysis for non-coding RNA sequences. Answer: Thank you for your suggestion. In our revised manuscript, GO analysis was used for annotation of target genes of non-coding RNAs. The manuscript is suggested to be proof-read by a native English speaker to improve its quality. I noticed a number of grammatical errors. Here below are what I found, as I am not a native English speaker, I cannot be sure I am right, but certainly it is worthy to have a look. Answer: Thank you for your suggestion, we revised our manuscript by an Australia partner according to your suggestion, and all modified content is presented in yellow background. Minor comments 1) Title: type should be types. A number of places, the term needs to be double checked. Such as line 11, etc. 2) Line 18: locueses - wrong work, should be loci. 3) Line 21: was “the” predominant type. “A” “the” or plural etc need to be checked through the whole paper. 4) Line 24-26: the sentence is not clear, need to re-write. 5) Line 33: isone – is one 6) Line 42: locating – should be “located” 7) Line 53: grows – should be “grow” 8) Line 62-63: affecting should be “affects” 9) Line 193: It is not a complete sentence, need to re-write 11) Line 310: “suffer” is not good term to use in this context. 13) Line 107: locus should be “loci” 14) Line 109 – 116: make sure the format for numbers is consistent, such as 27244 -> 27,444, etc 15) Linre 130: specially, do you mean “specifically”? Answer: Thank you for your reminding, all spelling and grammar mistakes you mentioned are revised and marked as red in our revised manuscript. 10) Figure 5(A): % number is more than 100%, looks like the numbers with % mixed up. Answer: In figure 5a, The detail number and % are all listed. We have redrawn the photo to help the reader distinguish these two sets of numbers. 12) Line 314-316: “for convenience” – not clear what you want to express. Answer: I deleted these two words.

Reviewer 2 Report

The comprehensive transcriptome analysis and comparison between different culm types is a significant achievement. However, the presentation of the results is not adequate and seems to be just a list of figures. For example, some figures are not explained in detai in the main textl; in Figure 5F, the predicted size and the actual size of the bands seem to be different; in FIgure 6A, it does not seem to be a meaningful figure, since it is natural that where there are many genes, there are many AS and so on. I appreciate the aim of the study, but the way the results are presented is flawed and I don't think the paper is good enough to merit publication in the IJMS. 

Author Response

The comprehensive transcriptome analysis and comparison between different culm types is a significant achievement. However, the presentation of the results is not adequate and seems to be just a list of figures. For example, some figures are not explained in detail in the main text;

Answer: Thank you for your suggestion. In our revised manuscript, we add a brief language to introduce the purpose of our research work at the beginning of each section in the result and draw a succinct conclusion at the end of each section. Besides, we provided more detailed results in our revised manuscript.

In Figure 5F, the predicted size and the actual size of the bands seem to be different;

Thank you for your reminding, I mixed up the gel electrophoresis results of primer 1 and primer 2 for experimental verification result of PH02Gene08777.t1 in figure 5f. We corrected the order of gel electrophoresis results in our revised manuscript.

In Figure 6A, it does not seem to be a meaningful figure, since it is natural that where there are many genes, there are many AS and so on.

Thank you for your suggestion. We deleted figure 6A according to your suggestion.

I appreciate the aim of the study, but the way the results are presented is flawed and I don't think the paper is good enough to merit publication in the IJMS

Round 2

Reviewer 2 Report

The paper is improved. 

Author Response

Thank you for your kind suggestion for our study